# PD-L2 suppresses T cell signaling via coinhibitory microcluster formation and SHP2 phosphatase recruitment

Tomohiro Takehara[1,2], Ei Wakamatsu [2], Hiroaki Machiyama[2], Wataru Nishi [3], Katsura Emoto[4], Miyuki Azuma[5], Kenzo Soejima[1], Koichi Fukunaga[1] & Tadashi Yokosuka [2✉]

The coinhibitory receptor, PD-1, is of major importance for the suppression of T cell activation in various types of immune responses. A high-resolution imaging study showed that PD-1 forms a coinhibitory signalosome, "PD-1 microcluster", with the phosphatase, SHP2, to dephosphorylate the TCR/CD3 complex and its downstream signaling molecules. Such a consecutive reaction entirely depended on PD-1–PD-L1/2 binding. PD-L2 is expressed on professional antigen-presenting cells and also on some tumor cells, which possibly explains the discrepant efficacy of immune checkpoint therapy for PD-L1-negative tumors. Here, we performed precise imaging analysis of PD-L2 forming PD-1–PD-L2 clusters associating with SHP2. PD-L2 could compete with PD-L1 for binding to PD-1, occupying the same space at TCR microclusters. The PD-1 microcluster formation was inhibited by certain mAbs with functional consequences. Thus, PD-1 microcluster formation provides a visible index for the effectiveness of anti-PD-1- or anti-PD-L1/2-mediated T cell suppression. PD-L2 may exert immune suppressive responses cooperatively with PD-L1 on the microcluster scale.

[1] Division of Pulmonary Medicine, Department of Medicine, Keio University School of Medicine, Tokyo, Japan. [2] Department of Immunology, Tokyo Medical University, Tokyo, Japan. [3] Department of Thoracic Surgery, Graduate School of Medical Sciences, Kumamoto University, Kumamoto, Japan. [4] Department of Pathology, Keio University School of Medicine, Tokyo, Japan. [5] Department of Molecular Immunology, Graduate School of Medical and Dental Sciences, Tokyo Medical and Dental University, Tokyo, Japan. ✉email: yokosuka-ths@umin.ac.jp

T cell activation is generally modulated by two distinct signals, the first is antigen-specific signaling from the T cell receptor (TCR) and the second is antigen-independent signaling from various costimulatory and coinhibitory receptors. Programmed cell death 1 (PD-1, also known as CD279) is currently the most notable coinhibitory receptor to suppress T cell activation[1] and finely tunes various types of immune responses, such as immunotolerance, antiviral immunity, tumor immunity, and antibody maturation[2]. PD-1 is a member of the CD28 superfamily, which also includes CTLA-4[3], and inducibly expressed on T and B cells, natural killer cells, monocytes and dendritic cells[4] upon antigen or cytokine stimulation[5]. PD-1 contains two tyrosine motifs in its cytoplasmic tail, an immunoreceptor tyrosine-based inhibitory motif (ITIM) and an immunoreceptor tyrosine-based switch motif (ITSM)[6], that are phosphorylated possibly by a Src kinase, Lck, and/or C-terminal Src kinase upon the engagement of PD-1 in T cells[7]. Phosphorylation of both the ITIM and ITSM is required for the recruitment of Src homology 2 domain-containing tyrosine phosphatase 2 (SHP2), which contributes to the reduction of the overall phosphorylation status of the TCR/CD3–CD3ζ-associated protein of 70 kD (Zap70) axis and the CD28-protein kinase C-θ axis[6]. PD-1 possesses two ligands, programmed death-ligand 1 (PD-L1, B7-H1, or CD274)[8] and PD-L2 (B7-DC or CD273)[9], which belong to B7 family members. While PD-L1 is ubiquitously expressed throughout an entire body, notably by both immune cells and cancer cells[10], PD-L2 expression is relatively restricted to professional antigen-presenting cells (APCs) and increased when they are activated via innate receptor signaling[10].

Another PD-1 ligand, PD-L2 was recently reported to be expressed by cancer-associated fibroblasts[11] and various types of tumor cells[12]. The Cancer Genome Atlas and the results from the analyses of clinical tumor samples demonstrated that the expression of PD-L2 is sometimes more highly correlated with antitumor immune responses than PD-L1 in case of renal cell cancer and lung squamous cell cancer[13]. Practically, immunohistochemical staining of tumor tissues for PD-L2 is reported as a useful method to predict the efficacy of anti-PD-1 treatment in some kinds of malignancies[12]. Surface plasmon resonance analysis revealed that the affinity of PD-L2 for PD-1 is two-fold to six-fold higher than PD-L1[14]. The crystal structure analysis for human PD-1–PD-L2 complex suggested a possibility that this high-affinity binding can be the attractive target for the drug development with small compounds[15]. PD-L2 may possibly exert some influence over the developing tumor microenvironment by a different mechanism than PD-1–PD-L1 binding, therefore PD-L2 might be a promising newer target for antitumor immunotherapy. However, few studies have reported the biological and biochemical functions of PD-L2 for PD-1-mediated inhibition of immune cells and signaling. In particular, the differences between the two PD-1 ligands have remained elusive yet.

When a T cell recognizes its cognate antigen peptide in a major histocompatibility complex (pMHC) molecule expressed on an APC, an immunological synapse is formed at the adherent interface between the two cells[16]. By using the combined imaging system of high-resolution microscopy and antigen-presenting supported lipid bilayers (SLBs), we and other groups found that a few hundred clusters of TCRs are imaged at the immunological synapse[17–19]. These clusters consist of tens of TCRs and their proximal signaling molecules, many of which are tyrosine-phosphorylated, and they have been identified as a minimal unit for T cell activation and named TCR microclusters[19]. Similar to the TCR, tens of PD-1 molecules cluster together after binding their ligand, PD-L1, and form PD-1 microclusters at the same region of TCRs, accompanied by recruitment of the phosphatase SHP2. These PD-1–SHP2 microclusters contribute to the dephosphorylation of the TCR/CD3 complex and CD28, as well as their downstream signaling molecules[20].

In this study, we focused on the functional differences between PD-L1 and PD-L2 and examined PD-1-mediated T cell suppression from the viewpoint of PD-1 signalosome formation by using an advanced live cell imaging technique. We newly established SLBs reconstituted with both mouse (m) PD-L1 and mPD-L2 and confirmed the clustering of PD-1 triggered by PD-1–PD-L2 binding, which was blocked by several antibodies against PD-1, PD-L1, or PD-L2 in the specific manner to the antibody-antigen reactions. Similarly, the PD-1–PD-L2 binding transiently recruited SHP2, which induced dephosphorylation of TCR/CD3 complexes and their downstream signaling molecules at the TCR microclusters to suppress T cell responses. Our imaging system revealed the competition between PD-L1 and PD-L2 in binding to PD-1. When the density of PD-1 was less than its ligands, PD-L2 excluded PD-L1 from TCR-PD-1 microclusters, due to the higher affinity of PD-L2 for PD-1, while PD-L2 and PD-L1 colocalized together at the microclusters when there was a low density of PD-L2. Our study highlighted the significance of PD-L2 in the formation of the PD-1-mediated inhibitory signalosome, and also in the induction of immune suppression. These results could lead to an expanded panel of immune checkpoint inhibitors (ICIs) for a variety of clinical settings.

## Results

### PD-L2 induces PD-1 clustering at TCR microclusters in a manner similar to PD-L1.
PD-1 was previously reported to accumulate at an immunological synapse and also at TCR microclusters upon a T cell–APC conjugation[20]. To analyze the localization of PD-1 and its ligands more precisely, we utilized the SLB system, which contains glycosylphosphatidylinositol-anchored I-E$^k$ (I-E$^k$–GPI) and mICAM-1 as its basic components, to which mPD-1 ligands, mPD-L1 and/or mPD-L2, can be added. The molecular densities of mPD-L1–GPI or mPD-L2–GPI were calculated by flow cytometry, so that they could be adjusted to the same level expressed on a primary antigen presenting cell (Supplementary Fig. 1). Splenic CD4$^+$ T cells were prepared from AND-TCR [specific for moth cytochrome c 88–103 (MCC$_{88–103}$) on I-E$^k$] transgenic (Tg) Rag2-deficient (Rag2$^{-/-}$) PD-1-deficient (Pdcd1$^{-/-}$) mice, stimulated and then retrovirally transduced by enhanced green fluorescent protein-tagged mPD-1 (mPD-1–EGFP). The cells were allowed to settle on the SLB and imaged by confocal microscopy or total internal reflection fluorescence microcopy. On an SLB, PD-1 formed clusters at the nascent T cell-bilayer contact regions in the presence of PD-L1 or PD-L2 and the PD-1 microclusters migrated toward the center of an immunological synapse, forming a central-supramolecular activation clusters (c-SMAC) (Fig. 1a, b, and Supplementary Movie 1). To examine the colocalization of PD-1 with TCRs at microclusters, these AND-Tg CD4$^+$ T cells expressing mPD-1–EGFP were prestained with DyLight 650-labeled anti-TCRβ (H57) Fab and imaged on the same SLB as in Fig. 1a. In the presence of PD-L1 or PD-L2, PD-1 initially accumulated at the same clusters with TCRs (PD-1–TCR microclusters) (Fig. 1c, left panel) and later these PD-1 clusters eventually translocated into a c-SMAC (Fig. 1c, right panel and Fig. 1e, left). To compare the behaviors of PD-1 in cytotoxic T cells with those in helper ones, we imaged the splenic CD8$^+$ T cells from OT-I-TCR [specific for Ovalbumin 257–264 (OVA$_{257–264}$) on H2-K$^b$] Tg Rag2$^{-/-}$ mice on the same SLB except for H-2K$^b$–GPI, and confirmed no difference in the clustering and kinetics of PD-1 between the two T cell subsets (Fig. 1d, e, right).

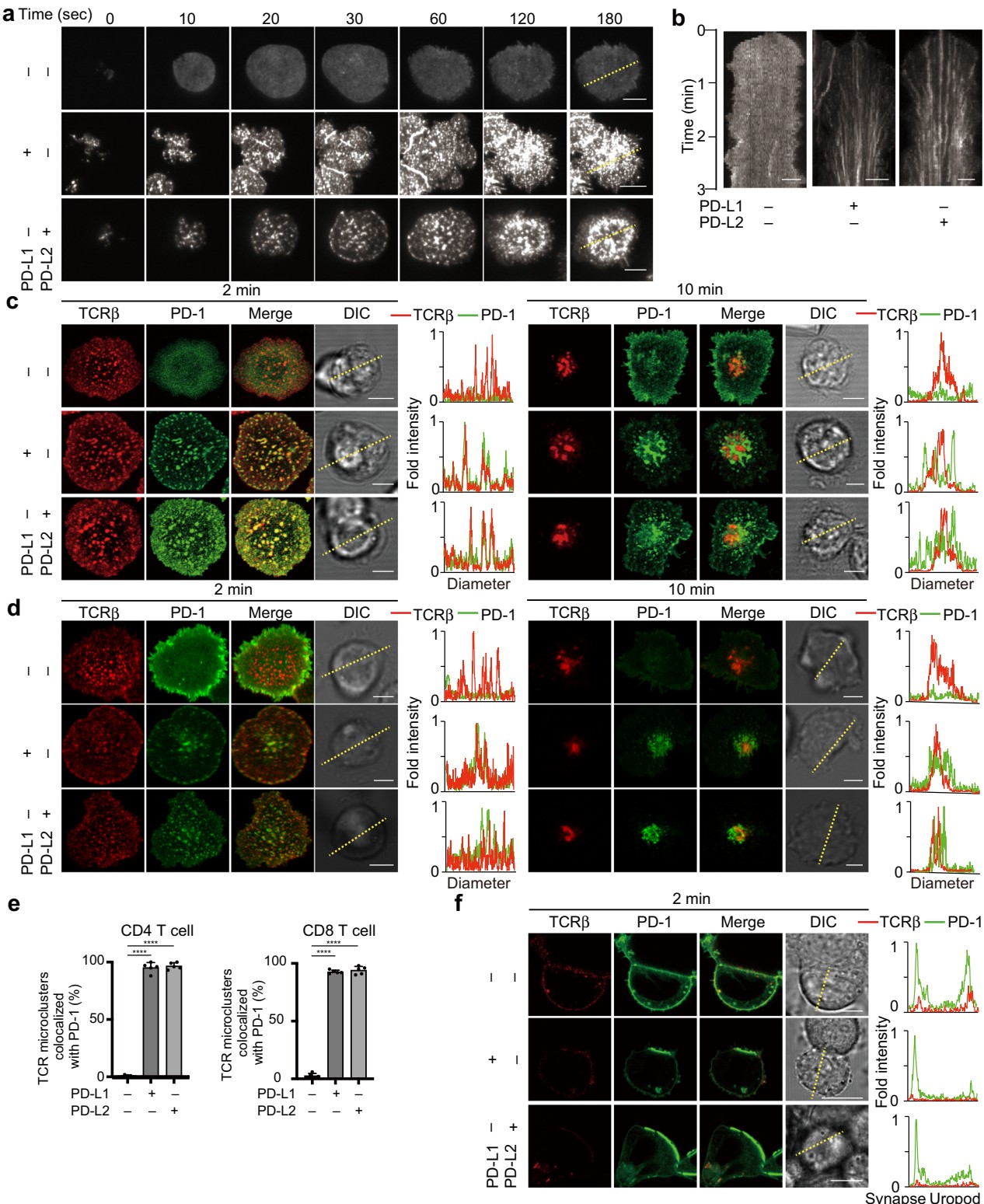

To examine the localization of PD-1 plus PD-L1 or PD-L2 at a T cell–bilayer interface, we conjugated a T cell expressing mPD-1–EGFP to an I-E$^k$-positive APC cell line, DC-1, introduced with mPD-L1 and/or mPD-L2 tagged by HaloTag. DC-1 cells were sorted based on the levels of staining for HaloTag to set PD-L1 and PD-L2 to all same density (Supplementary Fig. 2a–c). The accumulation of PD-1 could be imaged at an T cell–APC interface even if only PD-L2 was expressed (Fig. 1f). From these data, we confirmed that not only PD-L1 but also PD-L2 induces

both PD-1 microcluster formation, and its translocation to the c-SMAC associating with TCRs.

**PD-1–PD-L2 binding induces PD-1 microcluster formation colocalizing with both TCRs and CD28.** As shown in Fig. 1 with primary T cells, we confirmed PD-1 microcluster formation by using a T cell hybridoma expressing AND TCRαβ-chains (2D12) further introduced by mPD-1–EGFP. We observed PD-1 clusters

**Fig. 1 Translocation of PD-1 at TCR microclusters introduced by PD-1–PD-L1 or PD-1–PD-L2 binding. a** CD4[+] T cells were purified from AND-Tg *Pdcd1[−/−] Rag2[−/−]* mice, stimulated with irradiated B10.BR whole splenocytes with 5 μM MCC[88-103] peptides, and retrovirally transduced with mouse (m) PD-1–EGFP. The cells were plated onto an MCC[88-103]-prepulsed SLB containing I-E[k]–GPI (200/μm2) and mICAM-1-GPI (100/μm2) without (top) or with mouse mPD-L1-GPI (middle, 150/μm2) or mPD-L2-GPI (bottom, 150/μm2) and real-time imaged by total internal reflection fluorescence microscopy (times are above images; Supplementary Movie 1). **b** Clustering and centripetal movement of PD-1 on the diagonal yellow line in **a** is presented as horizontal elements in kymographs. **c** Primary CD4[+] T cells expressing mPD-1–EGFP (green) in **a** were prestained with DyLight 650-labeled anti-TCRβ (H57) Fab (red), plated onto an SLB as in **a** and real-time imaged by confocal microscopy at 2 (left) or 10 (right) min after contact. Histograms show fold fluorescent intensities of TCRβ (red) and mPD-1 (green) on the diagonal yellow lines in the DIC images. **d** CD8[+] T cells were purified from OT-I-Tg *Rag2[−/−]* mice, stimulated with irradiated C57BL/6 whole splenocytes with 100 nM OVA[257-264] peptide and retrovirally transduced with PD-1–EGFP. The cells were imaged as in **a** on an SLB containing OVA[257-264]-prepulsed H2-K[b]–GPI (200/μm2). Histograms are depicted as in **c**. **e** The graph shows the percentage of TCR microclusters colocalized with mPD-1 at 2 min after contact in CD4[+] T cells in **c** (left) and CD8[+] T cells in **d** (right) (*n* = 5). **f** AND-TCR T cell hybridomas (2D12) expressing mPD-1–EGFP were prestained with DyLight 650-labeled H57 Fab (red), conjugated with an MCC[88-103] prepulsed (5 μM) I-Ek-expressing APC line, DC-1 cell, not expressing (top) or expressing mPD-L1 (middle) or mPD-L2 (bottom) and real-time imaged by confocal microscopy at 2 min after contact. Histograms show fold fluorescent intensities of TCRβ (red) and mPD-1 (green) on the diagonal yellow lines in the DIC images. All data are representatives of three independent experiments. Bars, 5 μm. Error bars, SD. Statistical analysis was by one-way analysis of variance (ANOVA). ****p < 0.0001.

at the T cell–bilayer interface when mPD-L1–GPI or mPD-L2–GPI was reconstituted into the SLB (Supplementary Fig. 3a), and almost all of the T cells developed PD-1 clustering colocalized with TCRs (Supplementary Fig. 3b). Hui et al. reported that CD28 could be a primary target of PD-1-mediated inhibitory signaling[21]. CD28 possesses two ligands, CD80 and CD86, and augments both phosphatidylinositol-3 kinase–Akt and nuclear factor-kappa B signaling collaborating with TCRs after CD28–CD80/CD86 binding[22]. We showed the colocalization of PD-1 with both TCR and CD28 at microclusters by using the AND-TCR T cell hybridoma expressing mPD-1-HaloTag and mCD28–EGFP (Supplementary Fig. 3c). Although most of the PD-1, TCR, and CD28 microclusters were merged together, statistical analysis demonstrated that, in the presence of mPD-L1–GPI, there was a lower correlation efficiency between PD-1 and TCRs than those between PD-1 and CD28 or CD28 and TCRs (Supplementary Fig. 3d, left). In contrast, the correlation efficiency was almost identical between PD-1 and TCR, PD-1 and CD28, or CD28 and TCR in the presence of PD-L2–GPI (Supplementary Fig. 3d, right). These data indicated that PD-1 forms microclusters colocalized with both TCR and CD28 in the presence of CD80 and PD-L1 or PD-L2 with a slightly less tendency of PD-1–TCR co-clustering.

**Antibodies that block PD-1–PD-L1/2 binding interfere with PD-1 microcluster formation.** Anti-PD-1 and also anti-PD-L1 antibodies can have tremendous clinical benefits in treatment of human malignancies. To examine the effects of anti-mPD-1, anti-mPD-L1 and anti-mPD-L2 blocking antibodies on mPD-1 microcluster formation, we imaged the clustering of PD-1–EGFP on the AND-TCR T cell hybridoma settled onto the SLB in the presence or absence of antibodies against mPD-1 or mPD-L1 and/or mPD-L2. In the presence of mPD-L1–GPI on the SLB, PD-1 microclusters were disrupted by adding anti-mPD-1, anti-mPD-L1 or anti-mPD-L1 + anti-mPD-L2, but not by anti-mPD-L2 (Fig. 2a, left). If instead mPD-L2–GPI was on the SLB, anti-mPD-1, anti-mPD-L2 or anti-mPD-L1 + anti-mPD-L2, but not anti-mPD-L1, disrupted the PD-1 microclusters (Fig. 2a, right). The percentage of T cells forming PD-1 microclusters paralleled the blocking ability of each antibody (Fig. 2b). Various anti-mPD-1 clones have been already examined for their blocking efficiencies both in vitro and in vivo, therefore we verified available three typical clones of anti-mPD-1, J43[23], RMP1-14[24], and 29F.1A12[25], in our experimental settings. All clones worked properly as a staining antibody but they possessed their own optimal concentrations (Supplementary Fig. 4). We next compared the blocking efficiencies among these clones against the clustering of PD-1 in imaging. While J43 and RMP1-14

could not disrupted the clustering of PD-1 at a concentration of 10 μg/ml, they could at the higher concentration, 50 μg/ml. On the other hand, 29F.1A12 at a concentration of 10 μg/ml was enough to interfere the clustering of PD-1 (Fig. 2c). Collectively, these data showed that antibodies known as a blocking one can generally inhibit the clustering of PD-1 depending on their optimal concentrations.

**The phosphatase SHP2, but not SHP1, transiently associates with PD-1 microclusters triggered by PD-1–PD-L2 binding.** We next examined the association of the phosphatases, SHP1 and SHP2, to PD-1 microclusters generated by the PD-1–PD-L2 binding. By using AND-TCR T cell hybridomas co-expressing mPD-1-HaloTag and EGFP–mSHP1 or EGFP–mSHP2, we confirmed that SHP2, but not SHP1, colocalized at the PD-1 microclusters formed in the presence of either PD-L1 or PD-L2 at the very early phase after T cell-bilayer contact (20 s, Fig. 3a), and that SHP2 clustering could not detected at later time points. The translocation of SHP2 into the immunological synapse was further imaged at the interface between a T cell and an APC expressing PD-L1 or PD-L2 (Fig. 3b).

We also examined the physical association between PD-1 and SHP1 or SHP2 by western blotting using DC-1 cells expressing mPD-L1 or mPD-L2 (Supplementary Fig. 2a) prepulsed with MCC[88-103] peptides and AND-TCR T cell hybridomas expressing mPD-1–EGFP. Consistent with the above results, SHP2 was recruited to PD-1 upon the PD-1–PD-L1 or PD-1–PD-L2 binding, whereas the recruitment of SHP1 remained at background levels (Fig. 3c).

SHP2 is a responsible molecule in the PD-1-mediated T cell suppression and it could be a promising target to enhance the clinical effectiveness in ICI therapy. Meanwhile, SHP2 inhibitor was recently reported to suppress the in vivo growth of some kinds of cancers bearing *ras* mutation[26, 27]. We therefore examined the formation of PD-1-mediated coinhibitory microclusters in the presence of a SHP2 inhibitor, RMC4550. However, we could find any changes in neither clustering of PD-1 nor translocation of SHP2 (Supplementary Fig. 5), suggesting that physical association between PD-1 and SHP2 forming microclusters might not rely on the catalytic function of SHP2.

**PD-1–PD-L1/2 binding induces suppression of T cell activation by dephosphorylation of several signaling molecules in the downstream of TCR.** We next performed imaging analysis of the PD-1-mediated dephosphorylation of some of the signaling molecules, that are translocated into the TCR microclusters after

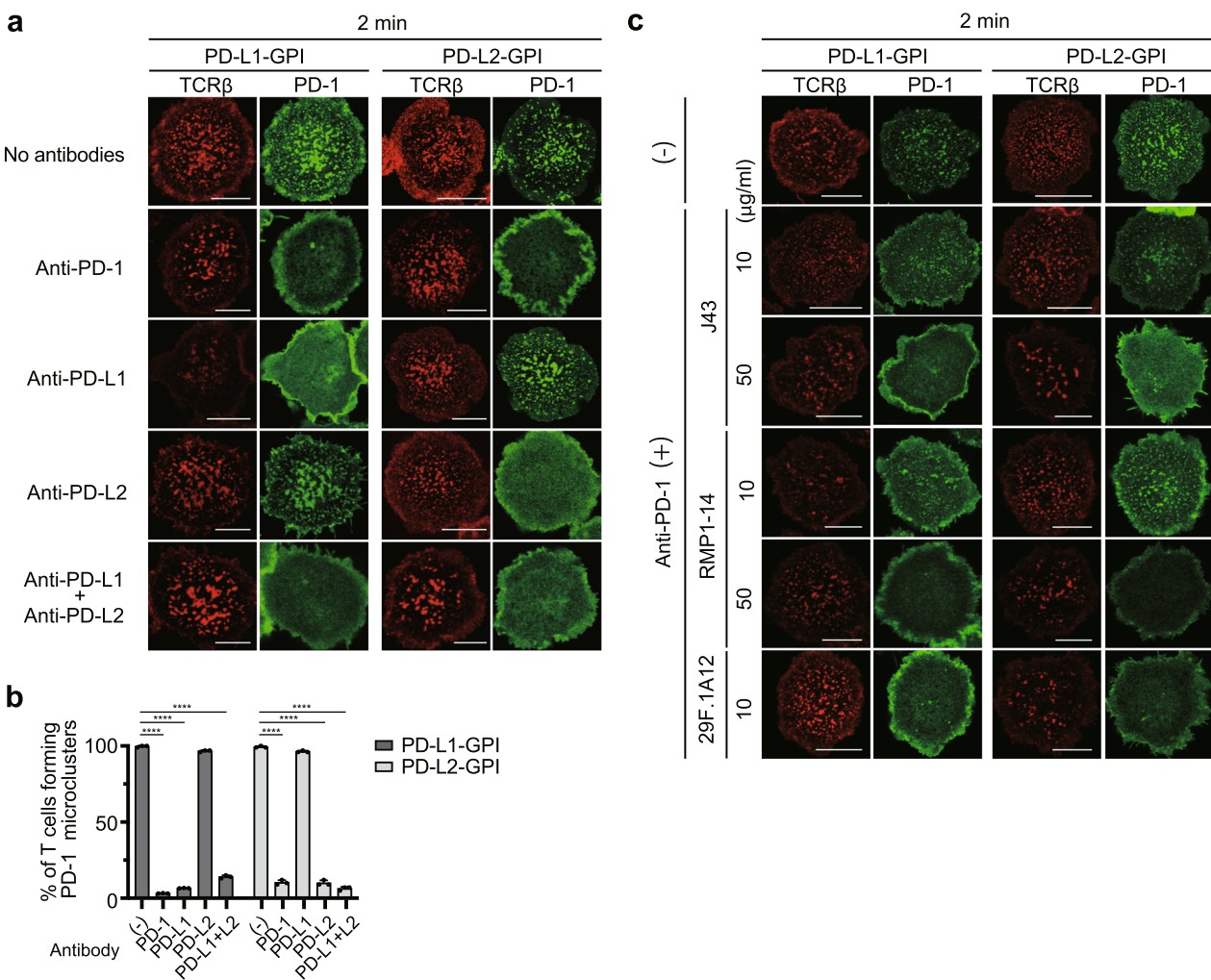

**Fig. 2 Each blocking antibody for PD-1, PD-L1, or PD-L2 requires its own concentration to block PD-1 microcluster formation. a** 2D12 expressing mPD-1–EGFP (green) cells were prestained with DyLight 650–labeled H57 Fab (red) and plated on an SLB with mPD-L1–GPI (left two columns) or mPD-L2–GPI (right two columns) as in Fig. 1a. The cells were real-time imaged by confocal microscopy at 2 min after contact in the absence (top) or presence of anti-PD-1 (29F.1A12, row 2), anti-PD-L1 (MIH5, row 3), anti-PD-L2 (MIH37, row 4) or both anti-PD-L1 and anti-PD-L2 (bottom). **b** The graph shows the percentage of T cells forming PD-1 microclusters in **a** ($n = 30$). **c** 2D12 expressing mPD-1–EGFP were imaged as in **a** in the absence (top) or presence of three different anti-PD-1 mAbs, J43 (row 2 and 3), RMP1-14 (row 4 and 5) or 29 F.1A12 (bottom) at a concentration of 10 or 50 μg/ml. All data are representatives of three independent experiments. Bars, 5 μm. Error bars, SD. Statistical analysis was by one-way analysis of variance (ANOVA). ****$p <$ 0.0001.

simultaneous binding of both TCR–pMHC and PD-1–PD-L1/2. The phosphorylation of CD3ζ was reduced at the TCR microclusters upon PD-1–PD-L1 or –PD-L2 binding (Fig. 4a). Quantification analysis clearly demonstrated that the ratio of phospho (p) CD3ζ staining to PD-1 (pCD3ζ/PD-1) decreased (Fig. 4b). The phosphorylation of CD3ζ and SLP-76 in primary CD4$^+$ T cells were also reduced at the TCR microclusters upon PD-1–PD-L1 or PD-1–PD-L2 binding (Fig. 4c, d). Phosphorylation of the molecules in the further downstream of TCR/CD3 complex, PLCγ1, Akt, and Erk, was also attenuated (Fig. 4e and Supplementary Fig. 6). We further confirmed the reduction of IL-2 production by mPD-1-expressing T cell hybridomas stimulated by DC-1 cells expressing, or not expressing mPD-L1 or mPD-L2 at any concentration of antigen peptides (Fig. 4f). Using these T cells, DC-1 cells and blocking antibodies as in Fig. 2a, we showed that each antibody restored IL-2 production by T cells, except for anti-mPD-L2 in the case of DC-1 cells expressing mPD-L1 and anti-mPD-L1 with DC-1 cells expressing mPD-L2 (Fig. 4g). Primary OT-I Tg CD8$^+$ T cells transduced with mPD-1 (Supplementary Fig. 7a) were further tested for the reduction of

IFNγ production after stimulation with H-2K$^b$-expressing tumor cell line, EL-4 cells, with or without mPD-L1 or mPD-L2 expression (Supplementary Fig. 7b). Both cytotoxicity and IFNγ production by OT-I Tg CD8$^+$ T cells were suppressed if these cells were stimulated by EL-4 cells expressing mPD-L1 or mPD-L2 and the suppression was reversed by addition of anti-mPD-1 (29F.1A12) (Fig. 4h, i). Related to the results in Fig. 2c evaluating the function of different anti-mPD-1 antibodies in clustering of PD-1, functional difference in each clone in vitro was evaluated from the viewpoint of recovery of IL-2 production. 29F.1A12 fully restored IL-2 production by T cells, whereas J43 and RMP1-14 could partially restored it (Fig. 4j). From these results, PD-L2 suppresses T cell function through dephosphorylation of TCR/CD3 and its downstream molecules in a similar fashion as does another PD-1 ligand, PD-L1.

**PD-L2 takes an advantage of binding to PD-1 against PD-L1.** A recent clinical report described the dual-expression of PD-L1 and PD-L2 by some of malignant tumors[12]. We analyzed the

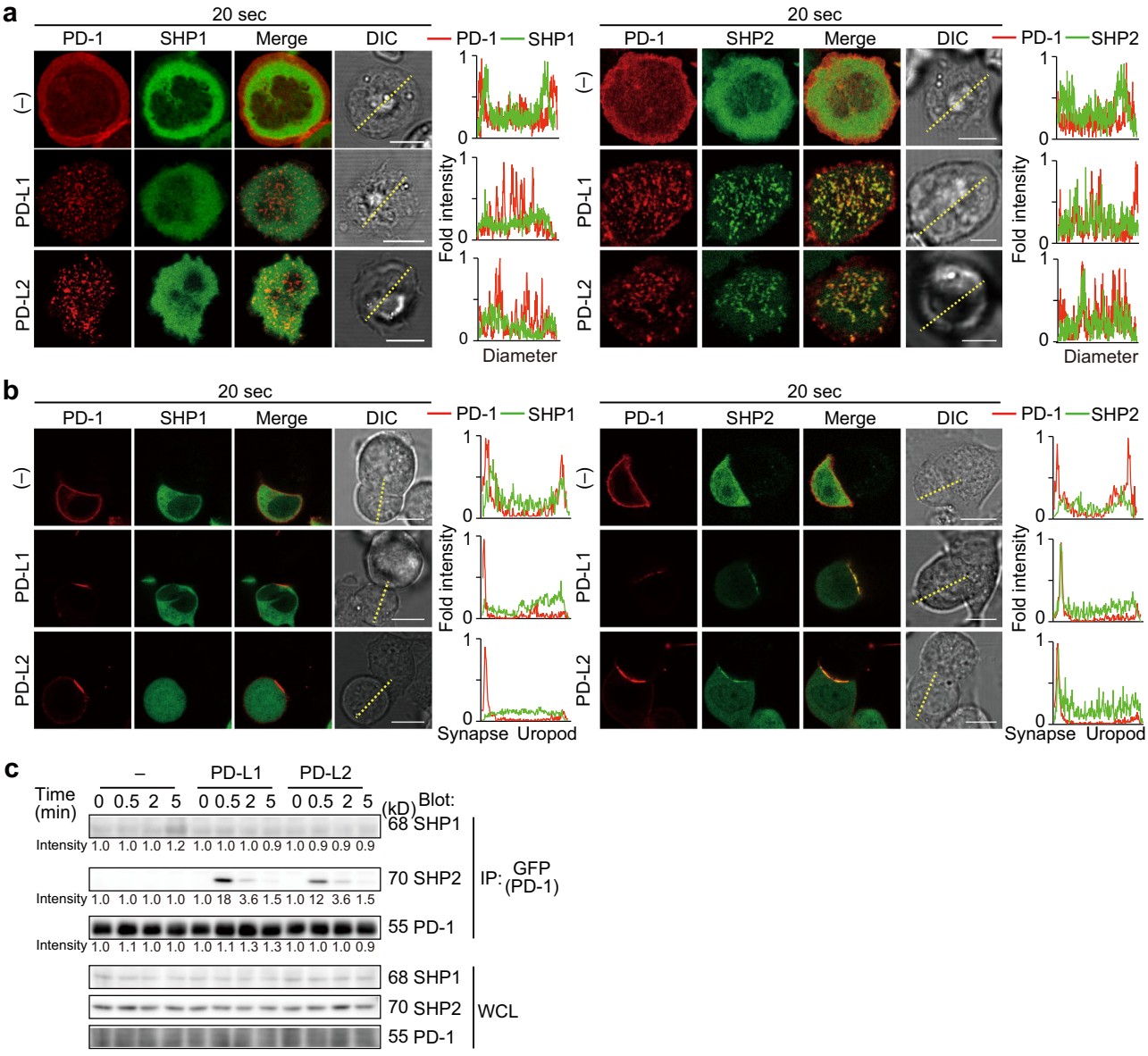

**Fig. 3 SHP2, but not SHP1, is rapidly and transiently recruited to PD-1 microclusters. a** 2D12 transduced with both mPD-1–HaloTag and EGFP–SHP1 (left) or EGFP–SHP2 (right) were preincubated with the HaloTag ligand–Stella650 (red), plated onto the SLB as in Fig. 1a without (top) or with mPD-L1–GPI (150/ μm², middle) or mPD-L2–GPI (150/μm², bottom) and real-time imaged by confocal microscopy at 20 s after contact. Histograms show fold fluorescent intensities of mPD-1 (red) and SHP1 (green) or SHP2 (green) on the diagonal yellow lines in the DIC images. **b** The cells in **a** (SHP1, left panel; SHP2, right panel) were conjugated with DC-1 cells not expressing (top) or expressing mPD-L1 (middle) or mPD-L2 (bottom) and real-time imaged by confocal microscopy at 20 s after contact. Histograms show fold fluorescent intensities of PD-1(red) and SHP1 (green, left panel) or SHP2 (green, right panel) on the diagonal yellow lines in the DIC images. **c** 2D12 expressing mPD-1-EGFP were conjugated by MCC$_{88-103}$-prepulsed (5 μM) DC-1 cells (left) or DC-1 expressing mPD-L1 (middle) or mPD-L2 (right) for the indicated times. The cells were lysed, immunoprecipitated for PD-1 by anti-GFP and blotted for SHP1, SHP2, or PD-1. The whole cell lysates (WCLs) were blotted for SHP1, SHP2, or PD-1. The number below each line represents the intensity of the band. All data are representatives of three independent experiments. Bars, 5 μm.

endogenous expression of human (h) PD-L1 and hPD-L2 by different human lung cancer cell lines before or after the culture with IFNγ, and found different expression patterns of PD-1 ligands in each lung cancer cell line (Supplementary Fig. 8). To mimic this expression pattern of PD-1 ligand, we reconstituted both mPD-L1–GPI and mPD-L2–GPI into an SLB, and confirmed the predominance of PD-L2 in binding to PD-1 with various densities of PD-L1 and PD-L2. As expected, PD-1 microclusters were still observed even at a low density of PD-L2, but not PD-L1 (Supplementary Fig. 9a, b).

We next examined the competition between PD-L1 and PD-L2 in the binding to PD-1. To analyze the distribution of PD-L1 and

PD-L2 on the same SLB, the ligands were labeled with different fluorescent dyes (PD-L1, AlexaFluor 647; PD-L2, AlexaFluor 488) and HaloTag-tagged PD-1 was stained by TMR-labeled HaloTag ligand after retroviral introduction into T cells. PD-L2 at a high density (more than 150 molecules/μm² in Fig. 5a, b, left, c) interfered with the translocation of PD-L1 within PD-1 microclusters, while PD-L2 still remained within PD-1 microclusters if PD-L1 was reconstituted even at a high density (300 molecules/ μm² in Fig. 5a, b, right, c).

We next attempted to evaluate T cell responses after stimulation by APCs with the same densities of PD-L1 and PD-L2 as those on the SLBs in Fig. 5a. We coated 5 μm silica beads

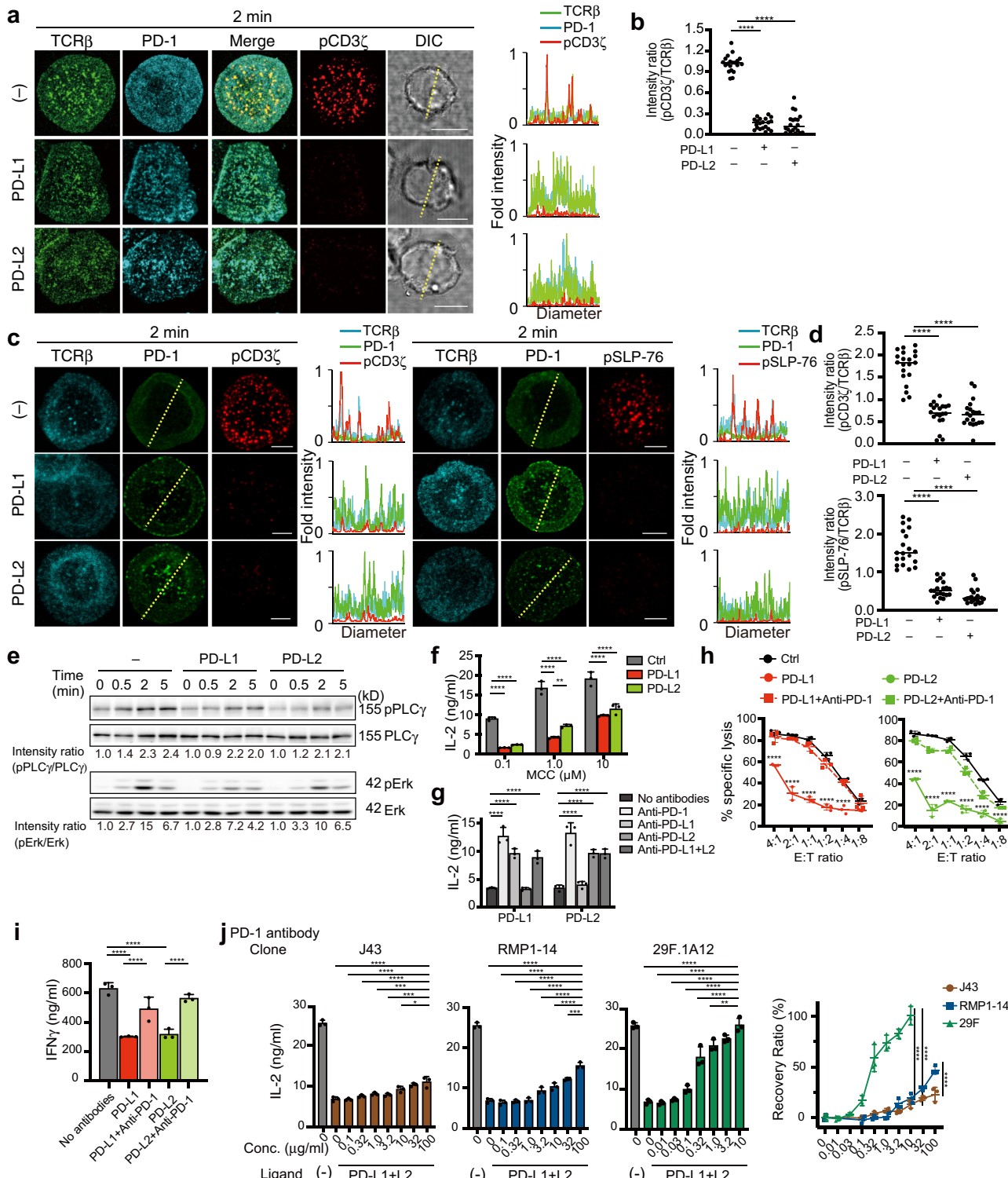

with the same lipid-bilayers as in Fig. 5a to mimic live APCs expressing these ligands. IL-2 production from T cells was suppressed, when the cells were stimulated by these silica beads with a fixed low-density mPD-L1–GPI (75 molecules/$\mu$m²) and mPD-L2–GPI with from high-density (300 molecules/$\mu$m²) to low-density. However, it was completely suppressed when the cells were stimulated by the beads with a fixed low-density mPD-L2–GPI (75 molecules/$\mu$m²) and mPD-L1–GPI from high to low (Fig. 5d).

We further tested the antigen-specific blocking and T cell suppression by immune checkpoint antibodies, anti-mPD-1, anti-mPD-L1, and/or anti-mPD-L2, in imaging. When a PD-1-expressing T cell settled on an SLB reconstituted with both mPD-L1–GPI and mPD-L2–GPI, PD-1 clustering was collapsed by adding anti-mPD-1 or anti-mPD-L1 together with anti-mPD-L2, but not by anti-mPD-L1 or anti-mPD-L2 alone (Fig. 5e). When these blocking antibodies were added into the conjugation between the T cells expressing PD-1 in Fig. 5e and DC-1 cells

**Fig. 4 PD-1–PD-L1 or PD-1–PD-L2 binding attenuates the phosphorylation of both TCR/CD3 complex and its downstream signaling molecules. a** 2D12 transduced with mPD-1–HaloTag were preincubated with the HaloTag ligand–TMR (cyan) and DyLight 488-labeled H57 Fab (green), plated onto the SLB as in Fig. 1a without (top) or with mPD-L1-GPI (150/μm$^2$, middle) or mPD-L2-GPI (150/μm$^2$, bottom), fixed at 2 min after contact, stained with Alexa Fluor 647-labeled anti-phospho (p) CD3ζ (red) and imaged by confocal microscopy. Histograms show fold fluorescent intensities of TCRβ (green), PD-1 (cyan) and pCD3ζ (red) on the diagonal yellow lines in the DIC images. **b** The graph shows pCD3ζ/TCRβ fluorescent intensity ratio at the T cell-bilayer interface in **a** in the absence (left) or presence of mPD-L1-GPI (middle) or mPD-L2-GPI (right) ($n = 20$). Horizontal bars, average. **c** Primary CD4$^+$ T cells in Fig. 1c transduced with mPD-1–GFP (green) were prestained with the DyLight 549-labeled H57 Fab (cyan), plated onto the SLB, fixed, stained for pCD3ζ (red, left panel) or pSLP-76 (red, right panel) and imaged as in **a**. Histograms show fold fluorescent intensities of TCRβ (cyan), PD-1 (green) and pCD3ζ or pSLP-76 (red) on the diagonal yellow lines in the middle column. **d** The graphs show fluorescent intensity ratio of pCD3ζ/TCRβ (top) or pSLP-76/TCRβ (bottom) in **c** in the absence or presence of mPD-L1-GPI or mPD-L2-GPI ($n = 20$). Horizontal bars, average. **e** 2D12 expressing mPD-1 were stimulated with MCC$_{88-103}$-prepulsed DC-1 cells not expressing (left) or expressing mPD-L1 (middle) or mPD-L2 (right) for the indicated times. The WCLs were blotted for pPLCγ1, PLCγ1, pErk1/2, or Erk1/2. The number below each line represents the intensity ratio, pPLCγ/PLCγ or pErk/Erk. **f** The cells in **e** were stimulated by 16h-aggregation culture with DC-1 cells (black) or those expressing mPD-L1 (red) or mPD-L2 (green) under the different concentrations of MCC$_{88-103}$ and the concentration of IL-2 in each supernatant was measured by ELISA. **g** The cells in **e** were stimulated by 16 h-aggregation culture with MCC$_{88-103}$ and DC-1 cells expressing mPD-L1 or mPD-L2 in the absence or presence of each antibody as in Fig. 2a and the concentration of IL-2 in each supernatant was measured by ELISA. **h** The target cell, EL-4 cell, is introduced by RLuc8 and further by mPD-L1 (red) or mPD-L2 (green). Effector primary CD8$^+$ T cells expressing mPD-1 were cocultured with 1 nM OVA$_{257-264}$-pulsed these target EL-4 cells at the indicated E:T ratios for 16 h with or without 10 μg/ml anti-PD-1 (29F.1A12). After treatment with RLuc8 substrate, the intensity of live cells was measured and the percent specific lysis was calculated. **i** The concentration of IFNγ in culture supernatants in **h** was measured by ELISA. **j** The cells in **e** were stimulated by 16 h-aggregation culture with 10 μM MCC$_{88-103}$ and DC-1 cells expressing mPD-L1 and mPD-L2 in the absence or presence of anti-PD-1 (J43: brown, RMP1-14: blue, 29F.1A12: green) as in Fig. 2c at the indicated concentrations and the concentration of IL-2 in each supernatant was measured by ELISA. The right graph shows the recovery rate of IL-2 production by each anti-PD-1 at the different concentrations. All data are representatives of three independent experiments. Bars, 5 μm. Error bars, SD. Statistical analysis was by one-way analysis of variance (ANOVA). *$p < 0.05$, **$p < 0.01$, ***$p < 0.001$, ****$p < 0.0001$.

expressing both mPD-L1 and mPD-L2, the accumulation of PD-1 at the immunological synapse was interfered by the addition of anti-mPD-1 or anti-mPD-L1 plus anti-mPD-L2 (Supplementary Fig. 10). Similarly, anti-mPD-1 or anti-mPD-L1 plus anti-mPD-L2 could restore, at least partially, the IL-2 production, whereas anti-mPD-L1 or anti-mPD-L2 alone could not (Fig. 5f). From these data, PD-L2 was shown to have relatively higher affinity than PD-L1 for PD-1 and was more efficient capacity for PD-1-mediated T cell inhibition.

## Discussion

Here, we have provided a comprehensive analysis of the inhibitory capacity of the less well-studied PD-1 ligand, PD-L2, as well as side-by-side analysis of activity of PD-L1 and PD-L2. PD-1–PD-L2 binding triggered the clustering of PD-1 with TCRs together forming a TCR–PD-1–PD-L2 signalosome in a fashion similar to PD-1–PD-L1 binding. Our studies, particularly the competition assays between PD-L1 and PD-L2 in the binding to PD-1, have resolved several issues of interest, scientifically and also clinically. First, upon binding to PD-L2, PD-1 forms coinhibitory microclusters colocalized with TCRs and transiently recruits the phosphatase SHP2. Second, we evaluated the blocking effects of several monoclonal antibodies by visualizing the collapse of PD-1 microclusters and confirmed the epitope-specific interference of each antibody. Third, PD-L2 occupies the space where PD-1 forms microclusters, preventing PD-L1 entry, when PD-1 was present at a relatively lower number than its ligands.

A TCR microcluster, the minimal unit for T cell activation, contains TCRs and their proximal signaling molecules such as Zap70, SLP-76, Vav1, and PLCγ1. We found that like PD-L1, PD-L2 induces clustering of PD-1 colocalized at a TCR microcluster in both CD4$^+$ and CD8$^+$ T cells. Immediate but transient recruitment of SHP2 to PD-1 microclusters was also the same when PD-1 was crosslinked by PD-L2. Since SHP2 translocates at a TCR–PD-1 signalosome, the substrates of PD-1-mediated dephosphorylation could be the downstream signaling molecules of TCR and also CD28. Based on the kinetics of the recruitment of SHP2 and the components of the signaling molecules within TCR–PD-1–SHP2 microclusters, PD-L1 and PD-L2 may convey similar inhibitory signals.

Surface plasmon resonance analysis has shown that PD-L2 can bind 2–6 times more potently to PD-1 than PD-L1[14]. Affinity between two molecules is generally measured as the interaction between an analyte and an immobilized ligand in surface plasmon resonance analysis, therefore it is not being evaluated as in a physiological situation, particularly in the case of receptor–ligand pairs both expressed on a cell surface. Our imaging analysis using SLBs thus has advantages for evaluating the affinity between two molecules on a cell surface correlating with their clustering. Each ligand on an SLB is movable and easily controlled in a density in a vast excess of lipids, therefore one can analyze the behavior of a single molecule in a spatiotemporal fashion. When clustering of PD-1 was imaged on an SLB reconstituted by PD-L1 or PD-L2 in a different density from high to low, PD-1 was tightly accumulated forming microclusters even in the presence of PD-L2 at a low density, indicative of the higher affinity of PD-L2 for PD-1. To make the density of PD-L1 or PD-L2 on SLBs close to the actual density on tumor cells, we adjusted their numbers and performed several near physiological, biological and biochemical assays. PD-1 signaling triggered by either PD-L1 or PD-L2 inhibits to almost the same extent the amount of IL-2 and IFNγ cytokine produced, and the cytotoxic function of CD8$^+$ T cells. This suggests that final T cell output is regulated by the PD-1 expression level, not by ligand expression level or its affinity for the receptor. The biological response of T cells triggered by two PD-1 ligands might be less affected, if the T cells expressed a smaller amount of PD-1 than those of PD-1 ligands on APCs. On the other hand, if PD-1 was more highly expressed on T cells compared to its ligands on APCs, the difference in affinity between PD-L1 and PD-L2 for PD-1 could totally contribute to PD-1-mediated inhibition of T cells. It might mean a delicate balance of the costimulatory and coinhibitory receptor network, which is harmoniously regulated by the affinity and also the spatiotemporal expression pattern of each receptor and ligand.

TCRs and costimulatory and coinhibitory receptors cooperatively manage T cell reactivity and various downstream signaling are considered to be the substrates primarily targeted by PD-1. Our imaging system with SLBs precisely elucidated the

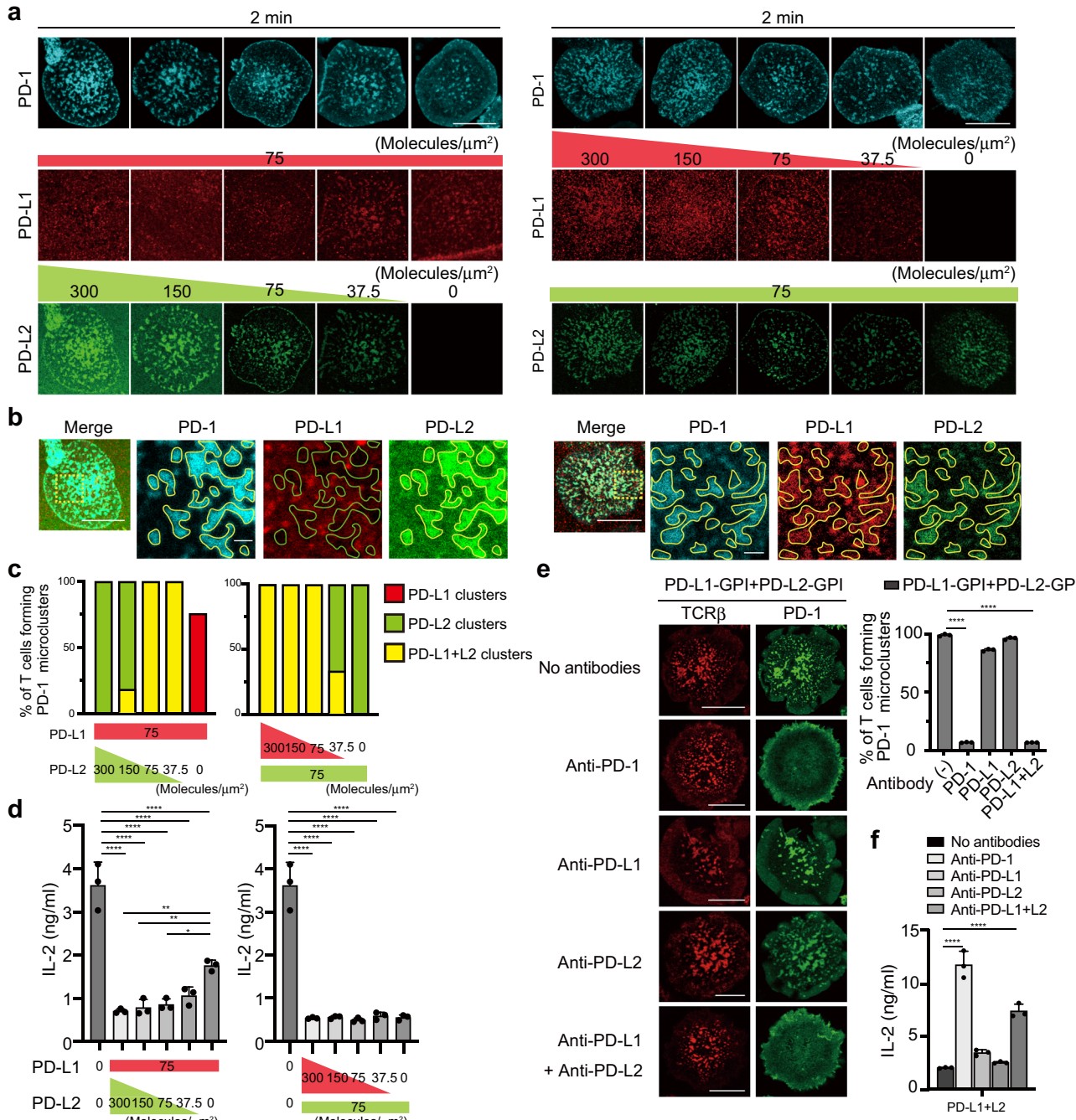

**Fig. 5 PD-L2 outcompetes PD-L1 in binding to PD-1 at the TCR–PD-1 microclusters. a** The mPD-L1–GPI (red, middle) and mPD-L2–GPI (green, bottom) were labeled with AlexaFluor 647 and AlexaFluor 488, respectively, and reconstituted into SLBs as in Fig. 1a in the different densities indicated above the images. 2D12 expressing mPD-1–HaloTag (top, cyan) were preincubated with HaloTag ligand–TMR and real-time imaged by confocal microscopy at 2 min after the T cell-bilayer contact. **b** Images of cells in **a** at mPD-L1–GPI (75 molecules/µm$^2$) + mPD-L2–GPI (300) (left) or mPD-L1–GPI (300) + mPD-L2–GPI (75) (right). The yellow squares in the left panels are magnified in the right three panels. Yellow enclosures show PD-L1 or PD-L2 clusters colocalized with PD-1 microclusters. Green enclosures show blank where PD-1 microclusters formed. **c** The graphs show the percentage of T cells forming PD-1 microclusters composed of either PD-L1, PD-L2, or PD-L1 + L2 at the different densities of mPD-L1–GPI and mPD-L2–GPI in **a**. **d** Silica beads were coated with the same SLBs as in **a** with different densities of mPD-L1–GPI and mPD-L2–GPI and prepulsed with MCC$_{88-103}$ to use as engineered antigen presenting cells. 2D12 expressing mPD-1 were stimulated by these silica beads for 16 h and the concentration of IL-2 in each supernatant was measured by ELISA. **e** 2D12 expressing mPD-1–EGFP (green) were prestained with DyLight 650-labeled H57 Fab (red) and real-time imaged on an SLB with both mPD-L1–GPI and mPD-L2–GPI in the absence or presence of each antibody as in Fig. 2a. The graph shows the percentage of T cells forming PD-1 microclusters ($n = 30$). **f** The cells in **e** were stimulated by 16h-aggregation culture with 10 µM of MCC$_{88-103}$ and the concentration of IL-2 in each supernatant was measured by ELISA. All data are representatives of three independent experiments. Bars, 5 µm. Error bars, SD. Statistical analysis was by one-way analysis of variance (ANOVA). *$p < 0.05$, **$p < 0.01$, ****$p < 0.0001$.

colocalization of PD-1 with both TCR and CD28 at microclusters. By statistical analyses, the correlation coefficient between PD-1 and TCR clustering was lower than that of PD-1–CD28, when PD-1 was crosslinked by PD-L1. In contrast, that of PD-1–TCR was almost same as that of PD-1–CD28 on the crosslinking of PD-1 by PD-L2. Recently, two papers reported that a *cis*-interaction of CD80 to PD-L1 inhibits PD-1–PD-L1 binding but still preserves the CD80–CD28 binding[28, 29]. Such differences in the correlation coefficient among PD-1, TCR, and CD28 clustering crosslinked by PD-L2 could be partly due to the *cis*-interaction between PD-L1 and CD80.

One of the mechanisms for antitumor effects of ICI treatment is the blocking of the inhibitory signaling via PD-1 resulting from PD-L1- or PD-L2-crosslinking. While anti-hPD-1 can block both PD-1–PD-L1 and PD-1-PD-L2 binding, the PD-1–PD-L2 binding still occurred even in the presence of anti-PD-L1. It is a critical point for the potential clinical efficacy of an ICI to evaluate whether the monoclonal antibody possesses a blocking effect, which results in interference between a receptor–ligand pair, and not a simple affinity against a receptor. This difference results from the location of the cognate epitope within an antigen. Anti-mPD-1 monoclonal antibodies, J43, RMP1-14, and 29F.1A12, can equally detect the surface expression of mPD-1 and they differently blocked the PD-1–PD-L1/L2 binding in our in vitro experiments (Fig. 2). Their function was almost consistent with the results of the imaging analysis for PD-1 clustering that J43 and RMP1-14 showed lower recovery ratio even with higher concentration of these antibodies compared with that of 29F.1A12. Although it is true that each clone functions differently in vitro and/or in vivo due to its isoform type or a real concentration in the microenvironment, our imaging technique can be effective to evaluate the actual function of the clones and to screen the most suitable one for ICIs. Anti-mPD-L2, which we examined in our in vitro and ex vivo experiments, fortunately restored both cytokine production and cytotoxicity against tumor cells expressing PD-L2. PD-1 is translocated into TCR microclusters together forming clusters upon crosslinking by PD-1 ligands and pMHC, and an antibody that blocks PD-1–PD-L1/2 binding collapsed PD-1 clustering but still maintained the TCR microclusters, which produce activation and restoration for T cell signaling. We found a tight correlation between the collapse of PD-1 microclusters and recovery of T cell functions, indicating the benefit of our imaging technique as a powerful screening tool for blocking antibodies likely to recover T cell functions more effectively.

CTLA-4 possesses two ligands, CD80 (also known as B7-1) and CD86 (also known as B7-2), whose expression pattern and affinity are different[30]. In a similar fashion as CTLA-4, while PD-L1 is known to be expressed on somatic cells constitutively and ubiquitously, PD-L2 is generally an inducible ligand. The purpose of having two ligands can be surmised as follows: PD-L1 is the main player maintaining peripheral tolerance by engaging PD-1 and that PD-L2 compensates for PD-L1 function, when the environment requires stronger inhibitory signaling through PD-1 due to, for example, excessive inflammation. Although PD-L1 is considered to be more predominantly expressed in various tumors, we are unfortunately aware of a biased usage of PD-L1 staining as a biomarker for a prospective ICI response. Recent papers have noted a much higher frequency of PD-L2-positive tumor cells than we expected. The expression of each PD-1 ligand on tumors differs with each organ and malignancy[31]. When we examined the expression of PD-L1 or PD-L2 on several human lung cancer cell lines before or after stimulation with IFNγ, those expression levels were shown to vary among the cell lines. A quite lower expression of PD-L2 was enough to form PD-1 microclusters and to introduce inhibitory signaling through PD-1, so anti-PD-1 should be prioritized. Anti-PD-1 is sometimes effective

in patients with non-small cell lung cancer, regardless of low PD-L1 expression, in which case it is a strong possibility that PD-L2 was highly expressed in the tumor mass.

In this paper, we have examined in detail the behavior of another PD-1 ligand, PD-L2, and elucidated similarities to the better-studied PD-1 ligand, PD-L1, its unique kinetics and their competition in binding to PD-1. Our results provide basic knowledge for a useful proposal of a new biomarker to indicate tumor immunotherapies. Anti-PD-L2 treatment is expected to be used in the future for some types of tumors.

## Methods

**Reagents**. The following antibodies and reagents we purchased; PE-anti-mCD80 (16-10A1), PE-anti-I-A/I-E (M5/114.15.2), FITC-anti-mPD-L1 or PE-anti-mPD-L1 (MIH5[32]), FITC-anti-mPD-L2 or PE-anti-mPD-L2 (MIH37[33]), PE-anti-mPD-L2 (TY25), PE-anti-hPD-L1 (MIH1), PE-anti-hPD-L2 (MIH18), FITC-isotype or PE–isotype-matched control IgG, anti-IL-2 (JES6-1A12) and biotin-labeled anti-IL-2 (JES6-5H4) from eBioscience; anti-mPD-1 (J43 and 29F.1A12) from Bio X Cell; anti-mPD-1 (RMP1-14) from BioLegend; Alexa Fluor 647-labeled anti-pCD3ζ (K25-407.69) and Alexa Fluor 647-labeled anti-pSLP-76 (J141-668.36.58) from BD; SHP2 inhibitor (RMC4550) from Selleck; rabbit polyclonal anti-SHP1 (C-19) and anti-SHP2 (C-18) and goat polyclonal anti-PD-1 (E-18) from Santa Cruz Biotechnology Inc.; anti-IFNγ (RA-6A2), biotin-labeled anti-IFNγ (XMG1.2), anti-Erk (4695S), anti-pErk (4370S), anti-PLCγ (5690S), anti-pPLCγ (8713S), anti-Akt (4691S), anti-pAkt (4060S), and HRP-anti-rabbit IgG polyclonal Abs from Cell Signaling Technology; HRP-anti-mouse IgG polyclonal Abs from Cappel. A B cell hybridoma producing anti-CD28 (PV-1) was provided by R. Abe (Tokyo University of Science, Noda, Japan); anti-TCRβ (H57-597) by RT. Kubo (Cytel Corp., CA, USA); anti-I-E$^k$ (14-4-4) and anti-ICAM-1 (YN1/1.7.4) by ML. Dustin (Univ. of Oxford, UK). Alexa Fluor 647-labeled anti-rat IgG (H + L), Alexa Fluor 647-labeled anti-hamster IgG (H + L), DyLight 650 and 549 labeling kits from Thermo Fisher Scientific; HaloTag (HT) STELLA Fluor 650 and TMR ligands from Promega; MCC$_{88-103}$ (ANERADLIAYLKQATK), and OVA$_{257-264}$ (SIINFEKL) peptides from GenScript.

**Mice and cells**. The AND-TCR-Tg mouse was provided by Dr. SM. Hedrick (University of California San Diego, San Diego, CA), *Rag2*$^{-/-}$ mice by Dr. F. Alt (Boston Children's Hospital, Boston, MA); OT-I-TCR-Tg *Rag2*$^{-/-}$ mice by Dr. W. Health (University of Melbourne, Melbourne, Australia); *Pdcd1*$^{-/-}$ mice (RIKEN BRC), Mice were maintained in specific pathogen free conditions at Tokyo Medical University. All experiments were performed in accordance with a protocol approved by the Animal Care and Use Committee of Tokyo Medical University (H30-0044, H31-0065, and R2-0001). The DC-1 fibroblast cells expressing I-E$^k$ and ICAM-1 were provided by J. Kaye (Cedars-Sinai Medical Center Los Angeles, CA). PLAT-E, the retrovirus packaging cell line, was provided by G. Nolan (Stanford University, Stanford, CA). Human lung cancer cell lines HCC827, H1299, and H3255 were purchased from ATCC. BHK and EL-4 cells were purchased from ATCC. The T cell hybridoma expressing the AND-TCR (AND hybridoma) was established by cell fusion of activated AND-Tg CD4+ T cells with lymphoma cell line, BW5147[22].

**Plasmid construction**. EGFP-tagged mPD-1, mCD28, mSHP1, and mSHP2 were generated by PCR and subcloned into retroviral vector, pMXs[20] (Kindly provided by Dr. T. Kitamura, Univ. of Tokyo, Japan). Renilla luciferase (RLuc) 8 and HaloTag (Promega)-tagged mPD-1, mPD-L1, or mPD-L2 were generated by PCR and subcloned into the pMXs retroviral vector. pMXs-RLuc8 was constructed by PCR using Yellow Nano-lanterns (kindly provided by Dr. Y. Okada, Riken, Japan[34]) as a template.

**Primary cell culture and transduction**. Retroviral vectors were transiently transduced into a packaging cell, PLAT-E (provided by G. Nolan, Stanford University, Stanford, CA) using Lipofectamine 2000 (Invitrogen). The supernatants were concentrated 40-fold by centrifugation at 8000×*g* for 12 h. CD4+ T cells were purified and stimulated with 5 μM MCC$_{88-103}$ and irradiated spleen cells from B10. BR mice. CD8+ T cells were purified and stimulated with 100 nM OVA$_{257-264}$, 200 U/ml recombinant mouse IL-2 and irradiated spleen cells from B6 mice. At 1 day after stimulation, the cells were suspended in retroviral supernatant with 10 μg/ml polybrene (Sigma-Aldrich) and 200 U/ml mouse IL-2 and centrifuged at 2600 rpm for 90 min at 37 °C. On day 2 or later, the cells were sorted to obtain the populations with homogeneous fluorescence intensity and then were maintained in RPMI1640 medium containing 10% FCS and mouse IL-2.

**Microscopy**. The cells expressing the proteins tagged with GFP stained by fluorescent-labeled H57 Fab and/or TMR-labeled or Stella650-labeled HaloTag ligand (Goryo Chemical) were allowed to settle onto an SLB. The cells were pre-incubated with anti-PD-1/L1/L2 antibody for 30 min or with SHP2 inhibitor for

2 h. The phosphorylation of CD3ζ and SLP-76 were detected by fluorescent-labeled anti-pCD3ζ and pSLP-76, respectively[20]. A confocal laser scanning microscope (TCS SP8, Leica Microsystems) comprising a 63× oil-immersion objective lens, high sensitivity HyD detectors and 488, 561, and 633 nm laser lines was used. LAS X software (Leica, Germany) was used for image acquisition. A TIRF analysis system was set up on a conventional inverted microscope (Ti-LAPP, Nikon, Tokyo, Japan) equipped with a TIRF objective lens (Nikon), a scientific CMOS camera (ORCA flash 4.0, Hamamatsu photonics) and fiber-coupled 488 nm lasers. The exposure time was set 100 ms with 2.5 s-interval between time points. NIS-elements software (Nikon) was used for image acquisition and ImageJ software (NIH, Bethesda, MD, USA, RRID:SCR_003070) was used for image processing and final figure preparation.

**Planar bilayers**. The purification and fluorescent labeling of GPI-anchored proteins have been established according to the protocols[35]. BHK cells highly expressing mPD-L1–GPI or mPD-L2–GPI were established. mPD-L1–GPI and mPD-L2–GPI was purified from the lysates by affinity column with MIH5 and MIH37, respectively. The mPD-L1–GPI and mPD-L2–GPI were labeled with AlexaFluor 647 and AlexaFluor 488, respectively. The expression level of each GPI-anchored protein on the planar bilayer was quantified using silica beads with a diameter of 5 μm (Bangs Laboratories)[20]. The densities were calculated based on the standard beads, Quantum FITC-5 MESF (Bangs Laboratories), and adjusted to the approximate concentration by comparison with natural APCs: I-E$^k$, 250 molecules/μm$^2$; mICAM-1, 100/μm$^2$; mCD80, 25/μm$^2$; mPD-L1, 37.5–300/μm$^2$ and mPD-L2 37.5–300/μm$^2$. The planar bilayers were loaded with 10 μM MCC$_{88–103}$ in citrate buffer, pH 4.5, for 24 h at 37 °C, blocked with 5% nonfat dried milk in PBS for 1 h at 37 °C, and left to stand in the assay medium (Hepes-buffered saline) containing 1% FCS, 2 mM MgCl$_2$, and 1 mM CaCl$_2$.

**Image processing**. The size and fluorescence intensity of each region were examined in all images and measured by ImageJ. The fluorescence intensities were measured based on the raw imaging data with the following formula. [intensity of fluorescence at each spot on a diagram] – [minimal intensity of each fluorescence on the entire line]) / ([mean intensity of each fluorescence on the entire line] – [minimal intensity of each fluorescence along the entire line][20]. Pearson's colocalization coefficients (PCCs) were calculated from each fold intensities.

**T cell–APC conjugation assay**. DC-1 or DC-1 cells expressing mPD-L1-Halotag and/or mPD-L2-HaloTag were prepulsed with 5 μM MCC$_{88–103}$ overnight at 37 °C. 2D12 expressing mPD-1-EGFP were cultured with prepulsed DC-1 cells with or without 10 μg/ml anti-PD-1, PD-L1, PD-L2 or PD-L1 + PD-L2 antibody. 2D12 expressing mPD-1-HaloTag and EGFP-mSHP1 or -mSHP2 were cultured with prepulsed DC-1 cells. The conjugates were visualized by confocal microscopy.

**Immunoprecipitation and Western blotting**. DC-1 cells were prepulsed with 5 μM MCC$_{88–103}$ overnight at 37 °C and washed before the assay. 1–2 × 10$^6$ AND hybridomas or primary CD4$^+$ T cells transduced with 0.5–1 × 10$^6$ DC-1 cells not transduced or transduced with mPD-L1, mPD-L2, or mPD-L1 plus mPD-L2 tagged with HaloTag. The cells were lysed with the lysis buffer (50 mM Tris-HCl, 50 mM NaCl, and 5 mM EDTA) containing of 1% NP-40. WCLs or those immunoprecipitated by anti-PD-1 were blotted with anti-PD-1 (1:1000), anti-SHP1 (1:500), anti-SHP2 (1:1000), anti-PLCγ (1:1000), anti-pPLCγ (1:1000), anti-Akt (1:2000), anti-pAkt (1:1000), anti-Erk (1:1000), or anti-pErk (1:1000) as a first antibody and HRP–anti-rabbit IgG polyclonal Abs (1:10,000) or HRP–anti-mouse IgG polyclonal Abs (1:10,000) as a second one. Each intensity of band was calculated by ImageJ.

**Flow cytometry**. Every staining antibody was used at a concentration of 2.5 μg/ml. A cell sorter, H800 (Sony), was used for cell isolation and a cell analyzed, Canto II (BD). was used for analysis. Data were analyzed using FlowJo software (Tree Star, Ashland, Oregon, United States).

**APC-modeled silica beads**. Silica beads were coated with I-E$^k$–GPI, ICAM-I–GPI and mPD-L1–GPI and/or mPD-L2–GPI in different densities. The silica beads were prepulsed with 10 μM MCC$_{88–103}$ at 37 °C for 16 h, incubated with 1% non-fat skim milk for blocking and cocultured with AND TCR T cell hybridomas as APCs.

**T cell stimulation assay**. 2 × 10$^4$ AND hybridomas were stimulated with 2 ×10$^4$ DC-1 cells expressing mPD-L1 and/or mPD-L2 with MCC$_{88–103}$ at the indicated concentrations in the presence or absence of 10 μg/ml anti-PD-1, anti-PD-L1, anti-PD-L2, or anti-PD-L1 + anti-PD-L2. 4 × 10$^4$ AND hybridomas expressing PD-1 were stimulated by MCC$_{88–103}$ prepulsed silica beads. 2.5–5 × 10$^4$ OT-I T cells were stimulated with 1 × 10$^5$ EL-4 cells expressing mPD-L1 or mPD-L2 with 1 nM OVA$_{257–264}$ in the presence or absence of 10 μg/ml anti-mPD-1. The concentration of IL-2 or IFNγ was measured by ELISA at 16 h after stimulation. All experiments were performed in triplicate.

**CTL killing assay**. RLuc8-introduced EL-4 cells were used as a target cell. At the indicated E/T ratios, mPD-1-transduced OT-I T cells were cocultured with 1 nM OVA$_{257–264}$ pulsed 1 × 10$^5$ EL-4 cells expressing mPD-L1 or mPD-L2 for 16 h in the presence of absence of 10 mg/ml anti-mPD-1. After treatment with RLuc8 substrate (FUJIFILM Wako), the intensity of RLuc8 luminescence in live target cells was measured by LAS4000. All experiments were performed in triplicate.

**Statistics and reproducibility**. Data were presented as the mean ± SD. Statistical analysis was performed by the Student's t-test or one-way ANOVA using GraphPad Prism 9.0. The p-values <0.05 were considered to be significance. Reproducibility including biological independent sample sizes and replicates are stated in each figure legend.

**Reporting summary**. Further information on research design is available in the Nature Research Reporting Summary linked to this article.

## Data availability

All data supporting the conclusions included in the manuscript are available within the paper and its supplementary information. Source data for figures and graphs in the main text can be found in Supplementary Data. Full immunoprecipitation and western blot images are included in Supplementary Figs. 11 and 12.

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

## Acknowledgements

We thank Dr. Toshio Kitamura for pMXs retroviral vector, Dr. Masahiko Kuroda and Dr. Shinichiro Ohno for lung cancer cell lines, Dr. Peter D. Burrows for critical reading of this manuscript, Dr. Hiroyuki Yasuda, Dr. Hideki Terai, Hiroko Toyota, and Masae Furuhata for technical assistance and Mai Kozuka for secretarial assistance. This work was supported by JSPS KEKENHI (JP20J10564, T.T.), (JP25113725, JP15H01194, JP16H06501, JP17H03600, JP19K22545, JP20H03536, T.Y.), PRESTO (U1114011, T.Y.) from Japan Science and Technology Agency, the Takeda Science Foundation (T.Y.) and the Naito Foundation (4465-135, T.Y.).

## Author contributions

T.T., E.W., and T.Y. designed research; T.T., E.W., H.M., W.N., K.E. and T.Y. performed research; M.A. and T.Y. contributed new reagents; T.T. and H.M. analyzed data; E.W., K.S., K.F., and T.Y. supervised research; and T.T., and T.Y. wrote the paper.

## Competing interests

The authors declare no competing interests.
