## [Peer Review File · Communications Biology]

Reviewers' Comments:

Reviewer #1:

Remarks to the Author:

In this manuscript, Takehara et al reported the role of PD-L2/PD-1 interaction in regulating TCR signaling pathway. Using Supported Lipid Bilayer (SLB) and TIRF microscopy, the authors visualized formation of PD-1 or TCR microclusters after incorporating PD-L1 GPI and/or PD-L2 GPI into SLB or APC cell lines. The authors found that PD-1 microclusters and TCR microclusters are co-localized on SLB containing PD-L2 as well as PD-L1. In addition, they showed that antibody-mediated blockade of PD-1/PD-L1 or PD-1/PD-L2 interactions disrupted PD-1 microcluster formation. Mechanistically, the authors suggested that ligation of PD-1 and its ligands might recruit SHP2 to PD-1 microclusters and dephosphorylate TCR downstream molecules including TCR complex, PLC gamma and Erk. Lastly, this report showed that PD-1 preferentially bound to PD-L2 over PD-L1 when its ligands were given at the same concentration, suggesting that PD-L2 might have the higher binding affinity to PD-1 than PD-L1.

Overall, this manuscript is well-written and contains data with a high quality. Also, the study fits to a scope of the journal. Since PD-1 biology is now well-appreciated and relevant to therapeutic approaches, this manuscript has strength in significance of study. The weakness of the study is little conceptual advance and novelty. SHP2 recruitment to PD-1 was well-known mechanism and Tang and Kim already showed that PD-L2 has higher affinity to PD-1, compared to PD-L1 [1]. Therefore, to improve the manuscript and strengthen this weak point, the followings are recommended to address.

Major concerns:

- 1) In Figure 1, can the authors perform the same experiment with CD8 T cells? PD-1 is also expressed on regulatory T cells, a subset of CD4 T cells. To generalize the observation on SLB, primary CD8 T cells also should be tested.
- 2) In Figure 2, the authors used J43 and 29F.1A12 to disrupt PD-1 microclusters. The authors observed that PD-1 microclusters were collapsed specifically by 29F.1A12 and reasoned that 29F.1A12 and J43 behave differently due to difference in specificity. Can the authors provide more supporting evidence behind this? Also, can the authors use RMP1-14 in addition to these clones in Figure 2 and Figure 4?
- 3) In Figure 3 and 4, can the authors use SHP2 inhibitors to test if SHP2 mediated dephosphorylation may be involved in PD-L2 induced PD-1 microcluster formation?
- 4) In Figure 4, can the authors look at TCR downstream phosphorylation in primary T cells?
- 5) In Figure 3 and Figure 4, can the authors clarify any difference between PD-L1/PD-1 and PD-L2/PD-1 signaling pathway? The western blot analysis should be quantified to assess any quantitative difference between PD-L1 and PD-L2 signaling.
- 6) Akt pathway is one of the most known signaling pathways modulated by PD-1 ligation. The authors should measure pAkt and related protein phosphorylation.

Minor concerns:

- 1) Some of figures are missing statistical analysis.
- 2) The title is confusing. Does PD-L2 forms PD-1 microclusters by recruiting SHP2? There is no evidence of SHP2 recruitment as the causal event for formation of PD-1/PD-L2 microclusters in this manuscript.

Reference

1. Tang, S.; Kim, P.S. A high-affinity human PD-1/PD-L2 complex informs avenues for small-molecule immune checkpoint drug discovery. *Proc Natl Acad Sci U S A* 2019, 116, 24500-24506, doi:10.1073/pnas.1916916116.

Reviewer #2:

Remarks to the Author:

PD-1 has two ligands named PD-L1 and PD-L2. Antigen-presenting supported lipid bilayers (SLB) were reconstituted with similar levels of mouse PD-L1 or mouse PD-L2. CD4 T cells isolated from AND-Tg mice and transduced with PD-1-EGFP were then plated on such SLB and used to analyze by confocal and TIRF microscopy the clustering of PD-1 at the resulting immune synapse. PD-1 formed clusters at the nascent T cell-SLB contact regions in the presence of PD-L1 or PD-L2 and the resulting PD-1 microclusters migrated toward the center of the immunological synapse to form c-SMAC. Most PD-1 microclusters were colocalized with TCR microclusters. Using an I-Ek-positive APC cell line (DC-1) in which with PD-L1 or PD-L2 were tagged by HaloTag, PD-L1 and PD-L2 were capable of inducing PD-1-TCR microclusters and c-SMAC formation. Using a AND-TCR T cell hybridoma expressing PD-1-HaloTag and CD28-EGFP, PD-1 was found to colocalize with both TCR and CD28 at microcluster in presence of CD80 and PD-L1 or PD-L2. Antibodies that block either PD-1-PD-L1 or PD-1-PD-L2 binding were capable of preventing PD-1 microcluster formation. Using AND-TCR T cell hybridomas co-expressing mPD-1-HaloTag and EGFP-mSHP1 or -mSHP2, SHP2, but not SHP1, colocalized at PD-1 microclusters formed in the presence of either PD-L1 or PD-L2 20 s after the onset of T cell-bilayer contact and disappeared at later time points (not depicted). The phosphorylation of CD3z found in the TCR microclusters was strongly diminished upon PD-1/PDL1 or PD-1/PD-L2 binding and restored upon blocking PD-1/PDL1 or PD-1/PD-L2 with specific antibodies. Finally, considering that the affinity of PD-L2 for PD-1 is two- to six- fold higher than that of PD-L1, the authors showed that when PD-L1 and PD-L2 are co-expressed on SLB, PD-L2 is capable of outcompeting PD-L1 for binding to PD-1. In conclusion, this study shows that the neglected PD-L2 ligand has functional properties similar to the PD-L1 ligand, providing a rational basis for its targeting in cancer immunotherapy.

Specific comment

It will be interesting to have the authors speculating on the transient SHP-2 recruitment. If SHP2 is gone after 20 s of interaction (Fig. 3), it is surprising to see that at 2 min the CD3-zeta molecules are still dephosphorylated. I will have expected them to be re-phosphorylated due to the lack of PD-1:SHP2 pairs.

Reviewer #3:

Remarks to the Author:

This study is a follow-up research to the work published on JEM in 2012, in which it was firstly reported that PD-1 molecules form micro-clusters with TCRs upon binding to one of its ligand PD-L1 and recruited phosphatase SHP2, leading to dephosphorylation of TCR/CD3 complex and CD28. That study described how PD-L1/PD-1 pathway worked on single cellular/ molecular biological level. Continuously using their unique precise imaging analysis, this study has new focuses: 1) if engagement of PD-L2, the other ligand of PD-1, could also lead to PD-1/TCR microcluster formation and the suppression of TCR downstream signals; 2) functional differences between PD-L1 and PD-L2 on single cellular/ molecular biological level. The new study highlighted the significance of PDL2 in the formation of the PD-1-mediated inhibitory signalosome and also in the induction of immune suppression. This study brought some new inspiration to current PD-1/PD-L1 immunotherapy, especially for the indications with high PD-L2 expression in tumor microenvironment. In the study, experiments are well performed and manuscript are in well written.

One major and one minor question need to be addressed:

1. In figure 2a and 2c, three different clones of anti-PD-1 mAb (RMP1-30, 29F.1A12 and J43) were tested in preventing the formation of PD-1/TCR microclusters. There seem to be inconsistencies between their abilities to prevent PD-1/TCR microclusters and to block the interactions of PD-1

with the ligands. RMP1-30 was previously reported NOT blocking the interaction of PD-1 with the ligands (Eur. J. Immunol. 2005. 35: 1773–1785), while it could prevent the PD-1/TCR microcluster formation in figure 2a. In addition, both 29F.1A12 and J43 were reported to block PD-1/PD-L1 and PD-1/PD-L2 interactions and boost T cell responses in mouse tumor models, while here only 29F.1A12 but not J43 prevent the PD-1/TCR microcluster formation in figure 2c.

Can J43 in hand block the interactions of PD-1/PD-L1 and PD-1/PD-L2 (Blocking assay may help to test the quality of the Ab, especially J43)?

If Abs quality has no problem, the results may lead to an important question which is whether the prevention of PD-1/TCR cluster formation by anti-PD-1 treatment is dependent on the abolishment of PD-1/PD-L1 or PD-1/PD-L2 interaction. If yes, can the authors explain the inconsistency? If not, which mechanism (blocking interaction or preventing micro clusters formation) is more important to anti-PD-1 function in suppressing TCR downstream signals?

2. Figure 5a, only one image of a single cell under each PD-L1 and PD-1 density setting was shown. Statistical results (bar graphs showing percentages of T cells forming PD-1/PD-L1 or PD-1/PD-L2 microclusters) for each density settings should be added.

March 4, 2021

Dear Reviewers,

Thank you for your letter of January 5th for our submitted manuscript (COMMSBIO-20-3492-T) with comments. We are pleased to hear your kind encouragement to submit a revised manuscript. The following are the point-by-point response to the reviewers' comments.

Reviewer #1

Major concerns:

1. *In Figure 1, can the authors perform the same experiment with CD8 T cells? PD-1 is also expressed on regulatory T cells, a subset of CD4 T cells. To generalize the observation on SLB, primary CD8 T cells also should be tested.*

As the reviewer mentioned, PD-1 is expressed in various subsets of T cells and other immune cells. The expression of PD-1 in CD8 T cells is particularly important to discuss the phenotypes of exhaustion of the cells. We now prepared primary CD8 T cells from OT-I Tg Rag2^{-/-} mice and imaged the PD-1 microclusters in these T cells settled on another SLB expressing Class I, H-2K^b, and OVA peptide. As we expected, the clustering of PD-1 in CD8 T cells is almost the same as those in CD4 T cells. The imaging data are shown in Fig. 1d and 1e, right and sentences are added from page 6 line 18 to page 7 line 3 in the main text. Since the data of the biological response of CD8 T cells were already shown in Fig. 4h and 4i, the explanation was now moved to page 11 line 11-16 in the main text. Of course, regulatory T cells also express PD-1 as the reviewer pointed out, but we think this issue is out of this manuscript and will be an important subject in a future paper. PD-1 is known to contribute immune suppressive function by Treg cells and the development of Treg cells, but PD-1 directly inhibits the activation signaling in effector CD4 and CD8 T cells. PD-1 in Treg cells possibly demonstrate the same behavior in effector T cells, belonging to the CTLA-4 imaging in Treg cells in our previous paper (Yokosuka T *et al.*, *Immunity*, 2010).

2) *In Figure 2, the authors used J43 and 29F.1A12 to disrupt PD-1 microclusters. The authors observed that PD-1 microclusters were collapsed specifically by 29F.1A12 and reasoned that 29F.1A12 and J43 behave differently due to difference in specificity. Can the authors provide more supporting evidence behind this? Also, can the authors use RMP1-14 in addition to these clones in Figure 2 and Figure 4?*

We really thank for the meaningful suggestions. As the reviewer commented, we also gave a similar question to the discrepancy between our results and those in pervious papers. We this time purchased the antibody, clone RMP1-14, form BioLegend and compared the differences in the quality and the biological efficiencies among three anti-PD-1 clones, J43, 29F.1A12 and RMP1-14. We first examined the basic binding capacities to the mouse PD-1 molecule by flowcytometric analysis and depicted the results in Supplementary Fig.4. These data demonstrated the less working property in J43 and RMP1-14 compared to 29F.1A12 in the same concentration of antibody protein. In imaging analyses, we re-examined the blocking efficiencies in the different conditions further added by much

higher concentration of antibodies. In the first submitted manuscript, J43 was not shown to block the PD-1-mediated suppression with the concentration at 10 µg/ml. However, both J43 and RMP1-14 could be evaluated as the blocking antibodies with the higher concentration at 50 µg/ml in Fig. 2c and in page 9 line 1-10 in the main text. Fig. 4j is the new data for the recovery of IL-2 production from T cells stimulated by antigen peptide in the presence of anti-PD-1, J43, RMP1-14 or 29F.1A12, at the different concentrations depicted and explained from page 11 line 16 to page 12 line 3 in the main text. Both J43 and RMP1-14 could partially restore the T cell function *in vitro*, which is possibly consistent with the previous papers, but we noticed the dramatic effects of 29F.1A12 for the PD-1 blockade in such a biological experiment. Although we could not give a correct interpretation to explain these differences among the clones, we supposed that it depends on the differences between *in vitro* and *in vivo* situations, the isoform types, the real concentrations in the microenvironment and so on. The discussion was added in the main text, page 17 line 3-9.

3) *In Figure 3 and 4, can the authors use SHP2 inhibitors to test if SHP2 mediated dephosphorylation may be involved in PD-L2 induced PD-1 microcluster formation?*

SHP2 is the responsible for downstream molecule in the PD-1-mediated T cell suppression and it could be a promising target to enhance the clinical effectiveness in anti-PD-1 therapy. On one hand, SHP2 inhibitor was recently reported to possess a suppressive function in the *in vivo* growth of some kinds of cancers bearing *ras* mutation (Chen YN *et al.*, *Nature*, 2016, Nichols RJ *et al.*, *Nat Cell Biol*, 2018). We totally agree the reviewer's comment, so we purchased SHP2 inhibitor, RMC4550 (Selleck), and examined the PD-1 microcluster formation in the presence of the inhibitor. We could not find any differences in the clustering of PD-1 with or without the SHP2 inhibitor. We added the new data as Supplementary Fig. 5 and described it in page 10 line 8-14 in the main text. We demonstrated that the clustering of PD-1 depends on the PD-1–PD-L1 binding not on the cytoplasmic domain of PD-1 in our previous paper (Yokosuka T *et al.*, *J Exp Med*, 2012), and speculated that these inhibitors would be an allosteric inhibitor for enzyme active site not for the regulatory domain of SHP2, which is essential for the recruitment to PD-1 microclusters. We found that the SHP2 inhibitor enhances the T cell response stimulated by APCs not expressing or expressing PD-L1 or PD-L2. Since the SHP2 inhibitor increases the base line of T cell response, we could not resolve whether the inhibitor affect the PD-1-mediated SHP2, SHP2 in the downstream of TCR or both (data not shown).

4) *In Figure 4, can the authors look at TCR downstream phosphorylation in primary T cells?*

We really agree the reviewer's comment and recognize the importance of the downstream signaling of TCR and PD-1 on primary T cells. We prepared the CD4 T cells from AND-Tg *Pdcd1*^{-/-} mice and imaged the phosphorylation of both the receptor CD3ζ and its downstream molecule SLP-76 by intracellular staining with phosphoprotein-specific antibodies. The imaging data are newly depicted in Fig. 4c and 4d and mentioned in page 11 line 3-5 in the main text.

5) *In Figure 3 and Figure 4, can the authors clarify any difference between PD-L1/PD-1 and PD-L2/PD-1 signaling pathway? The western blot analysis should be quantified to assess any*

quantitative difference between PD-L1 and PD-L2 signaling.

To answer the reviewer's comments, we quantified the results of Fig. 3c and 4e. We could not find a distinct difference between PD-L1/PD-1 and PD-L2/PD-1 inhibitory signaling to dephosphorylate the downstream of the TCR.

6) Akt pathway is one of the most known signaling pathways modulated by PD-1 ligation. The authors should measure pAkt and related protein phosphorylation.

We totally agree the importance of PI3K-Akt pathway in TCR signaling, which is also known to be a possible major target for PD-1-mediated inhibition, so we performed the Western blotting by primary AND-Tg T cells to confirm that the dephosphorylation status of Akt is attenuated in the presence of either PD-L1/PD-1 or PD-L2/PD-1 binding. The data is depicted in Supplementary Fig. 6 and explained in page 7 line 17-18 and page 11 line 5- 6 in the main text.

Minor concerns:

1) Some of figures are missing statistical analysis.

To follow the reviewer's recommendations, we added the statistical analysis to every data in Fig. 1e, 2b, 4d, 4g, 4h, 4i, 4j, 5d, 5e and 5f, and Supplementary Fig. 9b.

2) The title is confusing. Does PD-L2 forms PD-1 microclusters by recruiting SHP2? There is no evidence of SHP2 recruitment as the causal event for formation of PD-1/PD-L2 microclusters in this manuscript.

We never describe the SHP2 recruitment to form the PD-1/PD-L2 microclusters. We changed the title as "PD-L2 preferentially forms coinhibitory microclusters with both PD-1 and TCRs and recruits the phosphatase SHP2 to suppress TCR signaling".

Reference

1. Tang, S.; Kim, P.S. A high-affinity human PD-1/PD-L2 complex informs avenues for small-molecule immune checkpoint drug discovery. Proc Natl Acad Sci U S A 2019, 116, 24500-24506, doi:10.1073/pnas.1916916116.

We really thank for the reviewer's indication and apologize for missing the paper to be referred. We added this paper as number 15 in the reference and described the results from the paper in the main text, page 4 line 3-4.

Reviewer #2

Specific comment

It will be interesting to have the authors speculating on the transient SHP-2 recruitment. If SHP2 is gone after 20 s of interaction (Fig. 3), it is surprising to see that at 2 min the CD3-zeta molecules are

still dephosphorylated. I will have expected them to be re-phosphorylated due to the lack of PD-1:SHP2 pairs.

We understand and totally agree with the reviewer's comments and we feel difficulties in the biochemical analysis for very early T cell signaling. As the reviewer mentioned, the recruitment of SHP2 into the TCR/PD-1 microclusters is so quick within 20-30 seconds after the T cell-bilayer contact. But in our progressing study, we found that the initial phosphorylation of the TCR/CD3 complex arises as a much earlier event than PD-1. The kinase Lck is well known to phosphorylate CD3 ζ , the critical kinase ZAP-70 and PD-1. If we visualized Lck from the beginning of the T cell-bilayer contact under the single molecule level with high-spec TIRF microscopy, Lck is imaged to form a tiny clustering, so-called "nanoclusters", with TCRs and to leave the TCRs at the timing faster than the PD-1/SHP2 clustering. We believe that the absence of the Lck nanocluster at 20 s after the T cell-bilayer contact could explain the reason why CD3 ζ are not re-phosphorylated after the dissociation of SHP2 from PD-1/TCR microclusters. We apologize for presenting the imaging data of Lck as a Related Manuscript File for reviewers and the editor not as a content in this manuscript, because these studies are progressing and far from PD-1.

Reviewer #3

One major question:

1. In figure 2a and 2c, three different clones of anti-PD-1 mAb (RMP1-30, 29F.1A12 and J43) were tested in preventing the formation of PD-1/TCR microclusters. There seem to be inconsistencies between their abilities to prevent PD-1/TCR microclusters and to block the interactions of PD-1 with the ligands. RMP1-30 was previously reported NOT blocking the interaction of PD-1 with the ligands (Eur. J. Immunol. 2005. 35: 1773–1785), while it could prevent the PD-1/TCR microcluster formation in figure 2a. In addition, both 29F.1A12 and J43 were reported to block PD-1/PD-L1 and PD-1/PD-L2 interactions and boost T cell responses in mouse tumor models, while here only 29F.1A12 but not J43 prevent the PD-1/TCR microcluster formation in figure 2c.

Can J43 in hand block the interactions of PD-1/PD-L1 and PD-1/PD-L2 (Blocking assay may help to test the quality of the Ab, especially J43)?

If Abs quality has no problem, the results may lead to an important question which is whether the prevention of PD-1/TCR cluster formation by anti-PD-1 treatment is dependent on the abolishment of PD-1/PD-L1 or PD-1/PD-L2 interaction. If yes, can the authors explain the inconsistency? If not, which mechanism (blocking interaction or preventing micro clusters formation) is more important to anti-PD-1 function in suppressing TCR downstream signals?

We really thank for the meaningful comments and suggestions. First, we apologize for the simple mistake of RMP1-30 instead of 29F.1A12. We totally agree with the reviewer's comments and have discussed the similar question for the discrepancy between our results and those in previous papers the reviewer referred. To address this issue, we newly purchased another antibody, clone RMP1-14 from BioLegend, and compared the differences in the quality and the biological efficiencies among three anti-PD-1 clones, J43, 29F.1A12 and RMP1-14. We first examined the basic binding capacities to the mouse PD-1 molecule by FACS analysis. These new data in Supplementary Fig. 4 demonstrated the lower binding property of J43 and RMP1-14 compared to 29F.1A12 in the same

concentration of the antibody. In imaging analysis, we re-examined the blocking efficiencies in the different conditions, from low to ultra-high concentration of the antibodies. In the first submitted manuscript, J43 was not shown to block the PD-1-mediated suppression with a concentration at 10 $\mu\text{g/ml}$. However, both J43 and RMP1-14 could be evaluated as the blocking antibodies with a higher concentration at 50 $\mu\text{g/ml}$ (Fig. 2c and page 9 line 1-10 in the main text). The recovery of IL-2 production by adding J43, RMP1-14 or 29F.1A12 with the titrated concentrations depicted in Fig. 4j and explanation is added from page 11 line 16 to page 12 line 3 in the main text. We concluded that both J43 and RMP1-14 could partially restore the T cell function, which is possibly consistent with the previous papers. Although we also noticed the dramatic effects of 29F.1A12 in the PD-1 blockade, we could not give a correct interpretation to explain these differences among the clones. We supposed that it depends on the differences between *in vitro* and *in vivo* situations, the isoform types, the real concentrations in the microenvironment and so on, and add the sentences to discuss these discrepancies in the main text, page 17 line 3-9.

One minor question:

2. Figure 5a, only one image of a single cell under each PD-L1 and PD-1 density setting was shown. Statistical results (bar graphs showing percentages of T cells forming PD-1/PD-L1 or PD-1/PD-L2 microclusters) for each density settings should be added.

We really apologize for our incomplete analyses. We quantified the results of Fig. 5a and depicted the new bar graphs in Fig. 5c.

Revising the manuscript stated above, the original figure numbers are changed from Fig. 1d to Fig. 1e, from Fig. 1e to Fig. 1f, from Fig. 4c to Fig. 4e, from Fig. 4d to Fig. 4f, from Fig. 4e to Fig. 4g, from Fig. 5c to Fig. 5d, from Fig. 5d to Fig. 5e, from Fig. 5e to Fig. 5f, from Supplementary Fig. 4b to Fig. 4h, from Supplementary Fig. 4c to Fig. 4i, from Supplementary Fig. 1a to Supplementary Fig. 1a,b, from Supplementary Fig. 2a to Supplementary Fig. 2a,b,c, from Supplementary Fig. 4a to Supplementary Fig. 7a,b, from Supplementary Fig. 5a to Supplementary Fig. 8, from Supplementary Fig. 6 to Supplementary Fig. 9, from Supplementary Fig. 7a to Supplementary Fig. 10 and from Movie. 1a to Movie1. We newly added data availability in the text.

The figure number have been changed as summarized in below table.

Old version	New version
	1d
1d	1e
1e	1f
	4c
	4d
4c	4e
4d	4f
4e	4g
	5c
5c	5d

5d	5e
5e	5f
Sup. 4b	4h
Sup. 4c	4i
	4j
Sup. 1a	Sup. 1a, b
Sup. 2a	Sup. 2a, b, c
	Sup. 4
	Sup. 5
	Sup. 6
Sup. 4a	Sup. 7a,b
Sup. 5a	Sup. 8
Sup. 6	Sup. 9
Sup. 7a	Sup. 10
Movie. 1a	Movie. 1

In the light of above extensive revision by responding to all the reviewer's concerns, we hope that the revised manuscript is now acceptable for publication in *Communications Biology*.

Sincerely yours,

Tadashi Yokosuka, M.D., Ph.D.

Professor and Chairman,

Department of Immunology

Tokyo Medical University

6-1-1 Shinjuku, Shinjuku-ku, Tokyo 160-8402, Japan

Phone: +81-3-3351-6141

FAX: +81-3-3341-2941

E-mail: yokosuka-ths@umin.ac.jp

Reviewers' Comments:

Reviewer #1:

Remarks to the Author:

The authors properly addressed all the major and minor concerns that I suggested, therefore, the revised version of manuscript is significantly improved. The authors performed all of the suggested experiments to clarify the issues that were discussed in the initial review process. In conclusion, in my perspective, the new manuscript by Takehara et al is acceptable for publication.

Reviewer #2:

Remarks to the Author:

None

Reviewer #3:

Remarks to the Author:

The manuscript has been well revised. All my concerns were addressed.